# Rotation Plane Doubly Orthogonal Recurrent Neural Networks

**Zoe McCarthy, Andrew Bai, Xi Chen, & Pieter Abbeel**
Department of Electrical Engineering and Computer Science
University of California, Berkeley
Berkeley, CA 94720, USA
`{zmccarthy, xiaoyang.bai, c.xi, pabbeel}@berkeley.edu`

## Abstract

Recurrent Neural Networks (RNNs) applied to long sequences suffer from the well known vanishing and exploding gradients problem. The recently proposed Unitary Evolution Recurrent Neural Network (uRNN) alleviates the exploding gradient problem and can learn very long dependencies, but its nonlinearities make it still affected by the vanishing gradient problem and so learning can break down for extremely long dependencies. We propose a new RNN transition architecture where the hidden state is updated multiplicatively by a time invariant orthogonal transformation followed by an input modulated orthogonal transformation. There are no additive interactions and so our architecture exactly preserves forward hidden state activation norm and backwards gradient norm for all time steps, and is provably not affected by vanishing or exploding gradients. We propose using the rotation plane parameterization to represent the orthogonal matrices. We validate our model on a simplified memory copy task and see that our model can learn dependencies as long as 5,000 timesteps.

## 1 Introduction

Deep Neural Networks have shown increasingly impressive performance on a wide variety of practical tasks. Recurrent Neural Networks (RNNs) are powerful sequence modeling tools that have found successful applications in speech recognition, natural language processing, image captioning, and many more (Sutskever et al., 2014; Bahdanau et al., 2014; Wu et al., 2016; Donahue et al., 2015; Karpathy & Li, 2015; Luong et al., 2014). One fundamental problem with RNNs is the so called *vanishing and exploding gradients* problem (Hochreiter, 1991; Bengio, 1994; Hochreiter et al., 2001). These problems occur when training RNNs using gradient descent to model long sequences where the gradient magnitude either goes towards zero or infinity, respectively, as the length of the sequence increases.

Several heuristics have been proposed to alleviate these problems. The Long Short Term Memory (LSTM) in Hochreiter et al. (1997) and Gated Recurrent Units (GRU) in Cho et al. (2014) have been incredibly successful recurrent transition architectures to model complicated dependencies for sequences up to several hundred timesteps long and are the main RNN architectures in use today. The IRNN model modifies the standard RNN transition to initialize at the identity, which increases the timestep length modeling capability (Le et al., 2015). Stabilizing the forward hidden state norm can have a positive effect on hidden state gradient norm preservation (Krueger & Memisevic, 2015; Ba et al., 2016; Cooijmans et al., 2016). A simple gradient norm clipping during hidden state backpropagation can also help to alleviate the exploding gradient problem for these architectures (Graves, 2013).

Recently, there has been a surge of interest in orthogonal and unitary transition architectures. Orthogonal and unitary transitions exactly preserve forward and gradient norms passed through them. Theory developed for the linear neural networks suggests that training time can be vastly shorter when the weight matrices are orthogonal, and even independent of depth (Saxe et al., 2013). The Unitary Evolution Recurrent Neural Networks utilizes a unitary recurrent transition followed by a contractive nonlinearity to exactly solve the exploding gradient problem and greatly alleviate the

vanishing gradient problem (Arjovsky et al., 2015). A very recent extension expands on this work to increase the expressive power of these transitions and increases the validated architecture up to 2000 timestep dependencies in Wisdom et al. (2016). Analytic solutions to the most common long term dependency example tasks, the memory copy and addition problems, are provided with linear orthogonal recurrent neural networks in Henaff et al. (2016). A nonlinear activation function that is locally orthogonal is proposed in Chernodub & Nowicki and shown to have great potential. This particular activation scheme, however, is discontinuous (not just nondifferentiable), and could increase optimization difficulty in some cases.

An open question that we try to partially address in this work is whether a recurrent transition architecture can be fully orthogonal or unitary (and thus linear) and still learn expressive models to solve practical tasks. To address this problem, we propose the rotation plane doubly orthogonal RNN, a novel recurrent transition architecture that provably preserves forward hidden state activation norm and backpropagated gradient norm and thus does not suffer from exploding or vanishing gradients. The doubly orthogonal refers to the fact that the RNN transition architecture updates the hidden state multiplicatively by a time invariant orthogonal transformation followed by an input modulated orthogonal transformation. Rotation plane refers to the parameterization we use to represent the orthogonal matrices. We evaluate our approach on a simplified 1-bit version of the memory copying task, and find that our architecture can scale to 5,000 timesteps on this task, outperforming past approaches.

## 2 Doubly Orthogonal RNN Architecture

In this section we describe our proposed architecture, which we call the Doubly Orthogonal Recurrent Neural Net (DORNN). We also show that the DORNN provably preserves forward norm and gradient norm.

We briefly review the definition of an orthogonal or unitary matrix, since it is fundamental to the definition and properties of our transition architecture. Orthogonal matrices, or rotation matrices, are defined as matrices $Q \in \mathbb{R}^{n \times n}$ such that: $Q^T Q = I$. We restrict our attention to the special orthogonal matrices $SO(n)$ such that $\det(Q) = 1$. The set of special orthogonal matrices forms a matrix lie group. The complex analogue of an orthogonal matrix is a unitary matrix and is defined similarly as matrices $U \in \mathbb{C}^{n \times n}$ such that $U^H U = I$.

### 2.1 Recurrence Equations

Our recurrence equation is defined below:

$$h_{t+1} = R_{xh}(x_t) R_{hh} h_t$$

where $R_{xh}(x_t)$ is an input modulated orthogonal or unitary matrix, and $R_{hh}$ is a time invariant orthogonal or unitary matrix that is applied at every timestep. We parameterize $R_{hh}$ by $\theta_{hh}$ and $R_{xh}$ by $\phi_{xh}(x_t)$, where $\phi$ is a function of the input $x_t$. Figure 1 shows a graphical depiction of this transition architecture.

Our transition is different from most recurrent neural network transitions since it is fully multiplicative and contains no addition terms. The input enters into the equation by modulating the rotation applied to the hidden state. This allows for more expressivity in the hidden transition than a constant linear transition, though not as much as a nonlinear hidden state dependent transition. By having an input dependent transition, the hidden dynamics are more complicated than a constant linear transition and are nonlinear with respect to the input $x_t$. Linear time-varying models can express very complicated policies, such as in Levine et al. (2016). Our model can be viewed as a linear orthogonal time-varying model where the variation due to time is due to different input signals applied.

### 2.2 Forward Activation Norm and Gradient Norm Preservation

Here we prove that this recurrent transition architecture exactly preserves (up to numerical precision) the forward hidden state activation norm and the backwards gradient norm. All proofs carry forward

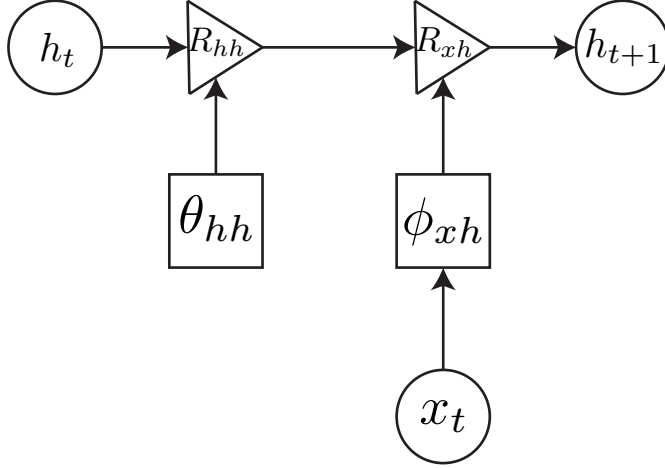

Figure 1: DORNN transition model. $R_{hh}$ and $R_{xh}$ are orthogonal transformations, parameterized by $\theta_{hh}$ and $\phi_{xh}$, respectively. The parameters $\phi_{xh}$ are a function of the input, $x_t$.

with unitary matrices instead of orthogonal matrices when transposes are replaced with Hermitian transposes.

$$||h_{t+1}|| = ||R_{xh}(x_t)R_{hh}h_t|| = ||R_{combined}(x_t)h_t|| = ||h_t||$$

where $R_{combined} = R_{xh}(x_t)R_{hh}$. The last equality follows since orthogonal matrices are a group and so $R_{combined} \in SO(n)$, and $||Qv|| = ||v||$ for any $Q \in SO(n)$ and any $v \in \mathbb{R}^n$, since $||Qv|| = \sqrt{v^T Q^T Q v} = \sqrt{v^T I v} = \sqrt{v^T v} = ||v||$ by the definition of $Q$ (orthogonal matrices preserve norm).

Now let $C(h_T)$ be a scalar cost function. The vanishing gradients problem occurs if $||\frac{\partial C}{\partial h_1}|| \to 0$ as $T \to \infty$ and the exploding gradient problem occurs if $||\frac{\partial C}{\partial h_1}|| \to \infty$ as $T \to \infty$.

$$\frac{\partial C}{\partial h_t} = \frac{\partial C}{\partial h_T}\frac{\partial h_T}{\partial h_t} = \frac{\partial C}{\partial h_T}\prod_{i=t}^{T-1}\frac{\partial h_{i+1}}{\partial h_i} = \frac{\partial C}{\partial h_T}\prod_{i=t}^{T-1}R_{hh}^T R_{xh}(x_i)^T$$

and so

$$\left|\left|\frac{\partial C}{\partial h_t}\right|\right| = \left|\left|\frac{\partial C}{\partial h_T}\prod_{i=t}^{T-1}R_{hh}^T R_{xh}(x_i)^T\right|\right| = \left|\left|\frac{\partial C}{\partial h_T}\right|\right|$$

where the last equality follows from $(\prod_{i=t}^{T-1} R_{hh}^T R_{xh}(x_i)^T) \in SO(n)$ : an orthogonal matrix's transpose is its inverse and the inverse of a group element is in the group. So the norm of the gradient of the cost $C$ at hidden state $h_t$ is the same as the final norm of the gradient at hidden state $h_T$, and the transition does not suffer from vanishing or exploding gradients.

## 3 ROTATION PLANE DOUBLY ORTHOGONAL RNN

Within the Doubly Orthogonal RNN, there is a choice in how the orthogonal (alternatively, unitary), transformations are parameterized. This choice determines the number of parameters, how the gradient propagates from the hidden state to the input, and much more. There are a wide variety of possible DORNN architectures, since there are a wide variety of different ways to parameterize orthogonal and unitary matrices, each with their pros and cons. We provide a particular instantiation of the Doubly Orthogonal RNN by parameterizing the orthogonal matrices in terms of the composition of many plane rotations. We call this RNN architecture the Rotation Plane Doubly Orthogonal RNN, or RP-DORNN.

We note that while we focus on this architecture within the context of a recurrent neural network, the rotation plane parameterization of orthogonal matrices could be equally useful for parameterizing very deep feedforward weight matrices.

### 3.1 ROTATION PLANE REPRESENTATION OF ORTHOGONAL MATRICES

First we show how we parameterize a single plane rotation. The full architecture is generated by a composition of a sequence of these plane rotations.

Well known in numerical linear algebra, any orthogonal matrix $Q \in SO(n)$ can be generated as the product of $n$ Householder reflection matrices $H = I - 2\frac{uu^T}{||u||_2^2}$ for nonzero vectors $u \in \mathbb{R}^n$.

Work in Geometric Algebra (for example see chapter 6 of Dorst et al. (2009)) gives us further insight into this parameterization. Two subsequent Householder reflections generate a plane rotation in the plane spanned by the reflection vectors. The generated rotation is twice the angle between the two reflection vectors. We can use this to parameterize rotation planes in term of the desired angle of rotation, by generating two reflection vectors that produce the rotation. By rotating in several planes in sequence, we can generate arbitrary orthogonal matrices. Thus we can view the rotation angle in a given plane as either a parameter to be optimized or as an input from below in the neural network, both of which we utilize in our proposed recurrent architecture.

Concretely, for a plane spanned by two orthonormal vectors, $w_0$, and $w_1$, we generate a rotation of angle $\theta$ from $w_0$ towards $w_1$ in the $w_0 - w_1$ plane by generating the following two reflection vectors $v_0$ and $v_1$ and composing their reflections.

$$v_0 = w_0$$

$$v_1 = \cos(\theta/2)w_0 + \sin(\theta/2)w_1$$

then $R_\theta = (I - 2v_1v_1^T)(I - 2v_0v_0^T)$. We don't need to divide by the magnitude of $v_0$ or $v_1$ since by construction they are unit vectors. When we apply $R_\theta$ to a vector or batch of vectors $B$, we don't have to generate the matrix $R_\theta$, since

$$R_\theta B = (I - 2v_1v_1^T)(I - 2v_0v_0^T)B = (I - 2v_1v_1^T)(B - 2v_0(v_0^T B)) =$$
$$= B - 2v_1(v_1^T B) - 2v_0(v_0^T B) + 4v_1(v_1^T(v_0(v_0^T B)))$$

and so we never have to form the full $R_\theta$ matrix, we only need to perform matrix multiplies of a vector with a matrix and the intermediate dimensions are much smaller than forming the dense $R_\theta$ matrix.

In the next section we treat $w_0, w_1$ as random constant orthonormal vectors and treat $\theta$ as a parameter or a function of inputs.

### 3.2 RP-DORNN

We generate $R_{hh}$ and $R_{xh}$ as a sequence of plane rotations in the Rotation Plane Doubly Orthogonal Recurrent Neural Network (RP-DORNN).

$$R_{hh} = \prod_{i=1}^{k} R_{\theta_i}$$

and

$$R_{xh}(x_t) = \prod_{i=1}^{l} R_{\phi_i(x_t)}$$

for $l, k \leq \lfloor \frac{n}{2} \rfloor$. Each $R_{\theta_i}$ and $R_{\phi_i}$ are plane rotations in randomized orthogonal planes (where the planes from $R_{hh}$ are orthogonal to one another and the planes from $R_{xh}$ are orthogonal to one another but the planes from $R_{hh}$ intersect with the ones from $R_{xh}$ randomly), parameterized by the angle of rotation. For $R_{\phi_i}$, the angle of rotation is a function of $x_t$, and for $R_{\theta_i}$. In our exploratory experiments we investigated several choices of $l$ and $k$ but for our final experiments we used $l = k = \lfloor \frac{n}{2} \rfloor$, so $R_{xh}$ and $R_{hh}$ can each affect the entire hidden space. For our experiments we generated the planes to rotate in for each of $R_{hh}$ and $R_{xh}$ by initializing a random Gaussian matrix and projecting to an orthogonal matrix, and taking consecutive pairs of columns as the orthonormal vectors that span a rotation plane ($w_0$ and $w_1$ for each plane in the notation of Section 3.1).

There are several possible choices for the parameterization of the angle parameters. In our exploratory experiments we investigated directly generating the angle value as a parameter but this suffers from topology wrapping: $\theta$ and $\theta + m2\pi$ for any integer $m$ generate the same transformation. We settled on generating $\theta_i = 2\pi\sigma(\alpha_i)$ for real parameters $\alpha_i$, where $\sigma$ is the sigmoid function and $\phi_i(x_t) = \pi\sigma(Wx_t + b)$ for learned affine transform $Wx_t + b$. This only allows positive angle rotations for the $R_{xh}$ and it places negative and positive angle rotations on the opposite side of the parameter space for $R_{hh}$ (negative $\alpha_i$ produces positive angle rotations and positive $\alpha_i$ produce negative angle rotations). In addition, with this parameterization, $\sigma(\alpha_i)$ and $\sigma(Wx_t + b)$ logarithmically approach 0 for very negative $\alpha_i$ and $Wx_t + b$, which is useful since for long timescales the transformations may be applied exponentially many times. In our experiments, $\alpha_i$ were drawn uniformly between $-3$ and $0$, $W$ was initialized with a Gaussian, and $b$ was initialized as $0$.

The Rotation Plane representation allows a fixed transformation subspace via the unvarying planes of rotation, but with a small number of angle parameters that are actively modulated to produce vastly different transformations for different inputs. Other orthogonal matrix representations do not give as much utility to be modulated by a small number of parameters that can be then generated by the input.

## 4    BACKGROUND: ALTERNATIVE ORTHOGONAL MATRIX REPRESENTATIONS AND OPTIMIZATION

### 4.1    UNITARY REPRESENTATION USED IN uRNN

The approach in Arjovsky et al. (2015) parameterizes their unitary matrices as a combination of unitary building blocks, such as complex reflections, complex unit diagonal scalings, permutations, and the FFT and iFFT.

### 4.2    OPTIMIZATION REPRESENTATION ON ORTHOGONAL MATRICES

Another way to parameterize an orthogonal matrix is to start with an arbitrary orthogonal matrix and then ensure that the optimization process produces an orthogonal matrix at each time step.

Several ways of optimizing orthogonal matrices. The defining equation can be used as a regularizer on a regular matrix transition, i.e. $||I - Q^T Q||$. This approach is used in some experiments in the ORNN paper, but it does not produce an exactly orthogonal matrix, and as such still suffers from exploding and vanishing gradients. Another possibility is to perform gradient descent on a matrix and reproject to the orthogonal matrices by SVD and setting the singular values to 1. The Lie algebra representation is $Q_{t+1} = Q_t q$ when $q = \exp(A)$ if $A$ is skew-symmetric: $A^T + A = 0$.

Then the representation optimizes $w$ in skew symmetric matrices. This approach is used in Hyland & Rätsch (2016). The main downside of this approach is that you need to calculate gradient through $\exp$, the matrix exponential, which we are not aware of a closed-form solution, and in Hyland & Rätsch (2016) the authors used finite differences to calculate the derivative of the exponential, which is incompatible with backpropagation. Recently proposed full capacity uRNN uses the Cayley transform: $Q_{t+1} = (I + \frac{\lambda}{2}A)^{-1}(I - \frac{\lambda}{2}A)Q_t$ for $A$ skew symmetric (skew-Hermitian for unitary $Q$) to stay on the manifold of orthogonal (unitary) matrices.

## 5 EXPERIMENTS

Our experiments seek to answer the following questions:

How do the theoretically guaranteed preservation properties of the DORNN architecture contribute to practical training to learn extremely long term dependencies, and how is training time to success influenced by the length of a long term dependency.

We partially address these questions on our RP-DORNN architecture on a simplified 1-bit version of the memory copy task that is commonly used to test long term dependencies in RNN architectures. The task is as follows:

The input is a sequence of four dimension one-hot vectors (categorical variables with four categories) of length $T + 2$ for a given $T > 1$. The first input is either a 1 or 2, followed by a string of $T$ 0's, and finally ends with a 3. All of the outputs except for the last output are arbitrary and unpenalized. The last output should be the same category as the first input (1 or 2, whichever was given). The loss is the categorical cross entropy of the last output timestep.

The simplifications from the usual memory copy experiment are that the number of categories is four instead of the usual ten, that the sequence to be copied is of length one instead of the usual ten, and that the loss only takes into account the portion of the output that corresponds to the copied output (the final timestep in our case), instead of the entire output sequence. These simplifications were performed in order to increase the number of experiments that could be run by training smaller models, and to function as a minimal experiment on the effect of increasing the sequence length. The third simplification was performed in order to increase the signal to noise ratio in the loss for very long sequences, since the majority of the loss comes from the intermediate outputs for very large $T$. The model would otherwise learn to use most of its capacity to reduce error in the irrelevant intermediate output stage instead of the relevant copied sequence stage. A useful side effect is that the entire gradient signal comes from the relevant copy output error.

We successfully trained the architecture on the task with $T = 600, 800, 1600, 2400, 3200$, and $5000$, with a minibatch size of 128. There are only two possible input and output sequences, so the minibatch size is somewhat redundant, but the different proportion of 1 and 2 inputs at each timestep helps inject extra randomness to the training procedure. Figure 2 shows the training loss curves for each of the timesteps, compared against the baseline of outputting 1 or 2 with equal probability on the final timestep, which achieves $ln(2) \approx 0.693$ loss. Figure 3 shows the accuracy of the network over training applied on the input minibatches. None of the hyperparameters were changed for scaling the timestep from 500 to 5000.

Note that for longer $T$, the training became increasingly unstable (although it was still able to succeed). A common phenomenon is that the loss would jump far above the baseline for a brief period but then return to a lower loss than before the jump. A possible explanation is that the training process becomes increasingly sensitive to small transition parameter changes as $T$ increases, since the transition is applied $T$ times, and so the final hidden state would land in a completely different region after a small parameter change and the network would have to adjust to the new regions.

Training was terminated for each $T$ after perfect accuracy had been achieved for several hundred training iterations, but the loss would continue to asymptotically approach $0$ if training were to progress as the network increases confidence in its predictions.

Saxe et al. (2013) suggests that training time should be independent of sequence length for a fully linear orthogonal neural network. We observed a roughly linear training time to success with respect to $T$. The discrepancy might be due to the nonlinearities present elsewhere in the network or due to the parameterization of the orthogonal matrices.

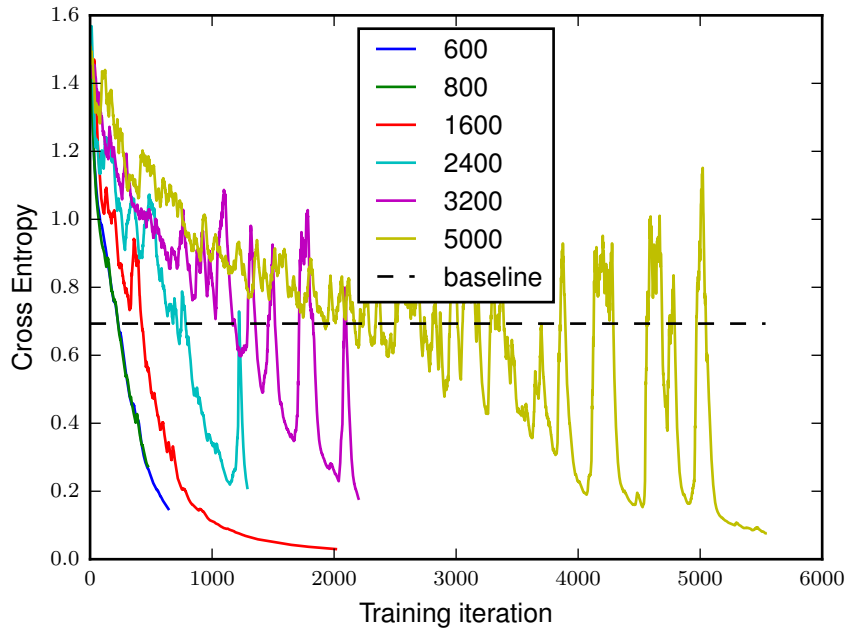

Figure 2: Exponentially weighted moving average (with decay of 0.95) of training loss.

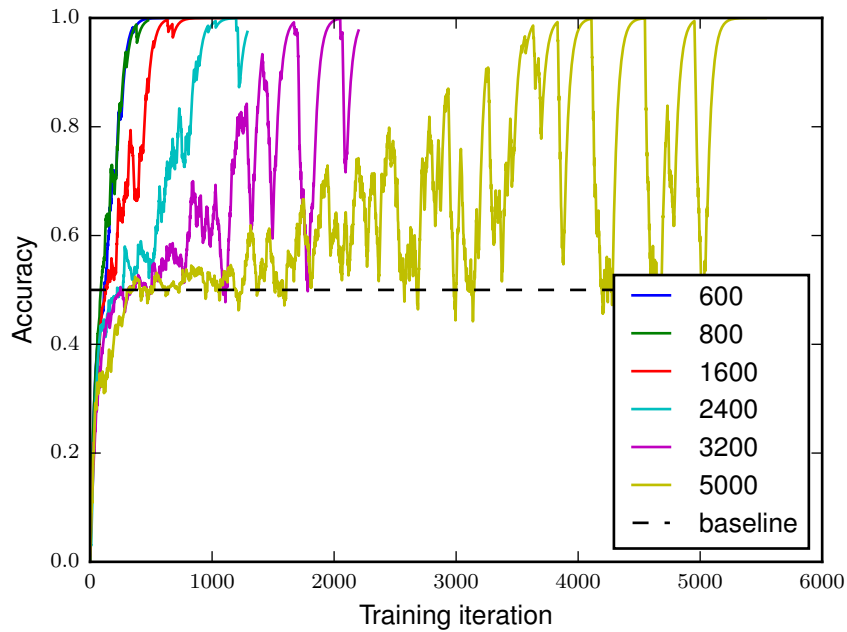

Figure 3: Exponentially weighted moving average (with decay of 0.95) of training accuracy.

This experiments section will be expanded upon as more experimental results and comparisons are available.

## 6 DISCUSSION

While we describe one particular orthogonal recurrent transition, the rotation plane parameterization describes a rich space of orthogonal transitions. Our input to hidden dependent transition is expressive enough for simplified memory copy task and can learn extremely long term dependencies up to 5,000 time steps long. The architecture still needs to be validated on more complicated tasks. Future work:

The architecture we described here is linear with the transition dependent on the input, but nonlinear locally orthogonal transitions are possible discontinuous. We experimented with different locally orthogonal transitions but found them harder to train successfully (likely they are more unstable due to the discontinuity, especially amplified over the hundreds or thousands of timesteps). It might be possible to find a nonlinear transition that still produces an orthogonal Jacobian that is continuous: the equations don't outright prevent it, but it results from the interplay of several constraints and so finding such an architecture is harder. An open question is whether is it possible to construct something like an LSTM within the space of orthogonal or unitary matrices, to get the best of both worlds? It is not clear how much expressivity in the recurrent transition is lost by having linear input dependent model instead of a nonlinear hidden to hidden transition.

A possible future direction is the combinations of orthogonal or unitary transitions for long term dependencies and regular RNN or LSTM units for complicated short term dependencies.

In this work we randomly fixed the planes to rotate in, but using the Cayley transform as in the full capacity uRNN the rotation planes could be optimized jointly, and so we could combine with our angle based parameterization to get full capacity input dependent rotations instead of being sensitive to the randomly initialized planes.

ACKNOWLEDGMENTS

This research was funded in part by Darpa through the SIMPlEX program and the FunLOL program, by NSF through a Career award. Zoe McCarthy is also supported by an NSF Graduate Fellowship. Peter Chen is also supported by a Berkeley AI Research (BAIR) lab Fellowship.

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
