# Peer review of "Rotation Plane Doubly Orthogonal Recurrent Neural Networks"

_ICLR 2017 — rejected_

[Official Review · AnonReviewer3 · rating 5 · confidence 3 · 15 Dec 2016]
**my review**

This is a nice proposal, and could lead to more efficient training of
recurrent nets. I would really love to see a bit more experimental evidence.
I asked a few questions already but didn't get any answer so far.
Here are a few other questions/concerns I have:

- Is the resulting model still a universal approximator? (providing large enough hidden dimensions and number of layers)
- More generally, can one compare the expressiveness of the model with the equivalent model without the orthogonal matrices? with the same number of parameters for instance?
- The experiments are a bit disappointing as the number of distinct input/output
sequences were in fact very small and as noted by the authr, training
becomes unstable (I didn't understand what "success" meant in this case).
The authors point that the experiment section need to be expanded, but as
far as I can tell they still haven't unfortunately.

[Official Review · AnonReviewer1 · rating 4 · confidence 4 · 17 Dec 2016]
**No Title**

This paper discusses recurrent networks with an update rule of the form h_{t+1} = R_x R h_{t}, where R_x is an embedding of the input x into the space of orthogonal or unitary matrices, and R is a shared orthogonal or unitary matrix.    While this is an interesting model, it is by no means a *new* model:  the idea of using matrices to represent input objects (and multiplication to update state) is often used in the embedding-knowledge-bases or embedding-logic literature (e.g. Using matrices to model symbolic relationships by Ilya Sutskever and Geoffrey Hinton, or Holographic Embeddings of Knowledge Graphs by Maximillian Nickel et al.).  I don't think the experiments or analysis in this work add much to our understanding of it.    In particular, the experiments are especially weak, consisting only of a very simplified version of the copy task (which is already very much a toy).  I know several people who have played with this model in the setting of language modeling, and as the other reviewer notes, the inability of the model to forget is an actual annoyance.   

I think it is incumbent on the authors to show how this model can be really useful on a nontrivial task; as it is we should not accept this paper.

Some questions:  is there any reason to use the shared R instead of absorbing it into all the R_x?  Can you find any nice ways of using the fact that the model is linear in h or linear in R_x ?

[Official Review · AnonReviewer2 · rating 4 · confidence 4 · 18 Dec 2016]
**not ready yet**

My main objection with this work is that it operates under a hypothesis (that is becoming more and more popular in the literature) that all we need is to have gradients flow in order to solve long term dependency problems. The usual approach is then to enforce orthogonal matrices which (in absence of the nonlinearity) results in unitary jacobians, hence the gradients do not vanish and do not explode. However this hypothesis is taken for granted (and we don't know it is true yet) and instead of synthetic data, we do not have any empirical evidence that is strong enough to convince us the hypothesis is true. 

My own issues with this way of thinking is: a) what about representational power; restricting to orthogonal matrices it means we can not represent the same family of functions as before (e.g. we can't have complex attractors and so forth if we run the model forward without any inputs). You can only get those if you have eigenvalues larger than 1. It also becomes really hard to deal with noise (since you attempt to preserve every detail of the input, or rather every part of the input affects the output). Ideally you would want to preserve only what you need for the task given limited capacity. But you can't learn to do that. My issue is that everyone is focused on solving this preserved issue without worrying of the side-effects. 

I would like one of these papers going for jacobians having eigenvalues of 1 show this helps in realistic scenarios, on complex datasets.

[Final Decision · Program Chairs · 06 Feb 2017]
**ICLR committee final decision**

Paper has an interesting idea, but isn't quite justified, as pointed out by R2. Very minimal experiments are presented in the paper.
 
 pros:
 - interesting idea
 
 cons:
 - insufficient experiments with no real world problems.
 - no rebuttal either :(.